# Expansion microscopy reveals subdomains in *C. elegans* germ granules

Kin M Suen[1] , Thomas MD Sheard[2], Chi-Chuan Lin[1], Dovile Milonaityte[1], Izzy Jayasinghe[2] , John E Ladbury[1]

**Light and electron microscopy techniques have been indispensable in the identification and characterization of liquid–liquid phase separation membraneless organelles. However, for complex membraneless organelles such as the perinuclear germ granule in *C. elegans*, our understanding of how the intact organelle is regulated is hampered by (1) technical limitations in confocal fluorescence imaging for the simultaneous examination of multiple granule protein markers and (2) inaccessibility of electron microscopy. We take advantage of the newly developed super resolution method of expansion microscopy (ExM) and in situ staining of the whole proteome to examine the *C. elegans* germ granule, the P granule. We show that in small RNA pathway mutants, the P granule is smaller compared with WT animals. Furthermore, we investigate the relationship between the P granule and two other germ granules, Mutator foci and Z granule, and show that they are located within the same protein-dense regions while occupying distinct subdomains within this ultrastructure. This study will serve as an important tool in our understanding of germ granule biology and the biological role of liquid–liquid phase separation.**

## Introduction

Germ granules, or nuage, are conserved features throughout the animal kingdom. Composed of conglomerates of RNP complexes, germ granules accommodate proteins and RNAs that play essential roles in epigenetics (Voronina et al, 2011). The P granule is a germ granule in *C. elegans*, and a number of Argonaute proteins responsible for siRNA pathways are found in this organelle (Seydoux, 2018; Sundby et al, 2021). For example, the piRNA pathway is a small non-coding RNA pathway that plays an important role in the regulation of transposable elements (Malone & Hannon, 2009; Weick & Miska, 2014; Czech et al, 2018), and the Argonaute protein PRG-1 (Piwi-related gene-1), which binds piRNAs, is localized to the P granule (Batista et al, 2008). Furthermore, disruption of P granules in embryos leads to misregulation of endogenous siRNA (endo-siRNA;

Dodson & Kennedy, 2019; Lev et al, 2019; Ouyang et al, 2019). Hence, the P granule is important for small RNA–based epigenetics.

The P granule was the first membraneless organelle (MLO) found to be formed by liquid–liquid phase separation (LLPS; Brangwynne et al, 2009). LLPS is a biophysical process by which biological molecules condense out of the bulk intracellular milieu to form distinct phases, allowing segregation of biological contents without the use of delimiting membranes (Banani et al, 2017; Wheeler & Hyman, 2018). Although the molecular rules that govern the formation of LLPS organelles have been intensively studied in the past decade, the implications, or the physiological roles, of LLPS in biological functions are difficult to assess. Further investigation of the spatial arrangements for biomolecules within phase separated MLOs can shed light on the role of LLPS in biological functions. For example, differences in translational activity between the periphery and the core of the P-bodies have been observed in Drosophila oocytes (Weil et al, 2012); surface tension was found to be responsible for the generation of the internal compartments within the nucleolus to facilitate ribosome biogenesis (Feric et al, 2016).

In this regard, the P granule presents particular challenges in understanding how LLPS facilitates its functions due to the complexity in its composition: more than 40 proteins have been identified as associated with this organelle (Updike & Strome, 2010; Suen et al, 2020; Sundby et al, 2021) and most RNAs in the germline are thought to at least associate with it transiently (Schisa et al, 2001; Sheth et al, 2010). Although certain proteins, for example, PGL-1, are used as markers to visualize P granules by proxy using fluorescence confocal microscopy, how certain treatments or mutations affect the P granule as a whole, or indeed whether subvariants of P granules exist, is difficult to appreciate relying on light microscopy. Furthermore, the lack of delimiting membrane makes it difficult to ascertain where the boundary of the organelle lies. In contrast, EM analysis of *C. elegans* germline tissue has played a critical role in characterizing the P granule without the bias use of specific protein markers. For example, Sheth et al showed that P granules are not homogeneous throughout but in fact consists of ultrastructures of a crest and a base and that the shape of P granules are dynamic (Sheth et al, 2010). However, the specific proteins that make up these ultrastructures are difficult to identify. Although methods such as immunogold staining and more recently

---

[1]School of Molecular and Cellular Biology, University of Leeds, Leeds, UK   [2]School of Biosciences, University of Sheffield, Sheffield, UK

Correspondence: j.e.ladbury@leeds.ac.uk; k.m.suen@leeds.ac.uk

correlative light and electron microscopy address this challenge (Pacy, 1990; de Boer et al, 2015), EM-based methods remain accessible only to specialised groups. This makes linking spatial organisations and physiological functions of the P granule difficult.

Given the aforementioned technical limitations of light and electron microscopy, and that protein arrangement within the germ granule appear to be organised on the nanometer scale (Wan et al, 2018; Putnam et al, 2019; Suen et al, 2020), we used the advanced super resolution technique protein retention expansion microscopy (proExM) coupled with indiscriminate proteome stains to visualise the germ granule. In proExM, proteins are chemically anchored to a swellable gel, which forms a mold of the fluorescently labelled sample. Proteins are non-specifically digested leaving a molecular-scale imprint of the staining covalently linked to the gel. Hydration of the gel allows gel to expand isotropically leading to the physical separation of the fluorophore and hence an increase in resolution in confocal imaging (Chen et al, 2015; Tillberg et al, 2016). Building on proExM, a whole-proteome staining method (i.e. a pan-protein stain) akin to densitometric stains used in EM was established, whereby protease digestion of the gel-anchored samples results in the decrowding of intracellular space while providing extra free amines for NHS ester fluorescent-labeling (M'Saad & Bewersdorf, 2020). This results in a protein density-encoded super resolution images that highlight cellular features such as plasma cell membrane, organelles, and spatial heterogeneities of intracellular protein organisation not restricted by the specificity of antibodies or targeted fluorescent probes.

To improve upon the resolution achievable with standard proExM in conjunction with confocal microscopy, we used the resolution-doubling Airyscan protocol (called enhanced expansion microscopy, EExM; [Sheard & Jayasinghe, 2021]) to image the proExM gels containing pan-protein staining of the bulk proteome in dissected *C. elegans* germlines. Among other intracellular features, we observed germ granules at nanoscale resolution (theoretical resolution ~40 nm in plane and ~100 nm axial). We used this EExM-pan stain pipeline to address two important aspects of germ granule biology: the effects of small RNA pathway mutations on germ granules, and the connectivity between different subtypes of germ granules: the P granule and two RNP granules, the Mutator foci and the Z granule.

## Results

Immunostaining of fluorescent proteins and pan-protein staining has recently been successfully carried out on whole mount *C. elegans* samples (Yu et al, 2020). However, perinuclear density in the germline that might be germ granules were not observed (Yu et al, 2020). We suspected this was due to the presence of the cuticle, and/or the fact that the germline was buried inside the body. proExM and EExM have been successfully performed on tissues without cuticles, as is the case for pan-protein staining for the bulk proteome. Given our interest is in the germline, we decided to test if the current pro/EExM and pan-protein stain protocols are suitable for *C. elegans* germlines extruded by dissection, which would manually remove the cuticle from the tissue of interest. Briefly, we dissected 1-d-old adults and used the freeze-crack

method to immobilise germline tissues on a coverslip throughout the immunostaining process. Immunostained germlines were then anchored to the swellable gel and digested. Before expansion, the gel was pan-protein stained with an NHS ester dye to label the bulk proteome. The gel was finally expanded and imaged using the super resolution mode on a confocal microscope (Fig 1A).

To determine the expansion factor achieved on the *C. elegans* germline tissue, we measured the diameter of the nucleus in the pachytene region via DNA staining using DAPI. We found that without expansion the nucleus diameter was 4.6 ± 0.4 $\mu m$ (mean ± SD; Fig 1B and C) whereas post-expansion nuclei were measured at 13.9 ± 2.0 $\mu m$. Hence, the expansion factor is ~three-fold on average. To ascertain whether isotropic expansion was achieved in the x- and y-planes, we sought to compare the germ granules before and after expansion. However, germ granules were not easily observed without expansion (Fig S1A). Therefore, we compared the aspect ratio of DAPI staining instead and found that expansion did not result in extortion in the plane (Fig S1B). We did not assess isotropy in the z-plane.

To aid the identification of germ granules, experiments were first performed on animals expressing GFP-tagged DEPS-1 (DEfective P granules and Sterile-1). DEPS-1 is a scaffold protein important for P granule assembly (Spike et al, 2008). Pan-protein staining in the pachytene region of the *C. elegans* germline proteome reveals a number of intracellular features including the plasma membrane, nucleus, and intranuclear structures (Fig 2A and Video 1). Perinuclear protein dense structures can be seen using pan-protein staining, which are similar to those observed previously with EM (Schisa et al, 2001; Sheth et al, 2010). Importantly, the co-localisation of DEPS-1 to these perinuclear structures confirms these are indeed germ granules (Fig 2A and B) and that the experimental methodology presented here allows the visualization of these. Using super resolution confocal microscopy, we previously showed that a number of protein condensates, including DEPS-1, are not homogenously distributed throughout the germ granule but organised as small protein clusters (Suen et al, 2020). EExM allows us to resolve DEPS-1 condensates further and confirms our previous observation. We determined the number of granules per nucleus in single optical sections using either DEPS-1 or pan-protein staining (Fig 2C). In most instances, perinuclear DEPS-1 staining coincides with perinuclear protein dense structures revealed by pan-protein staining as reflected by the similar average number (3.0 via DEPS-1 staining versus 2.9 via pan-protein staining) of granules observed in the optical sections examined.

### P granule, Mutator foci, and Z granule proteins are localised within the same germ granules

In addition to the P granule, three other perinuclear germ granules, the Z-granule (Ishidate et al, 2018; Wan et al, 2018), the Mutator foci (Phillips et al, 2012), and the SIMR foci (Manage et al, 2020), have been found in the *C. elegans* germline. Proteins important for various small RNA pathways in the germline often associate with these perinuclear granules. The functions that occur in these different granules/foci are sometimes interconnected. Hence, they are thought to be different condensates or subdomains within the same granule, rather than distinct granules (Sundby et al, 2021;

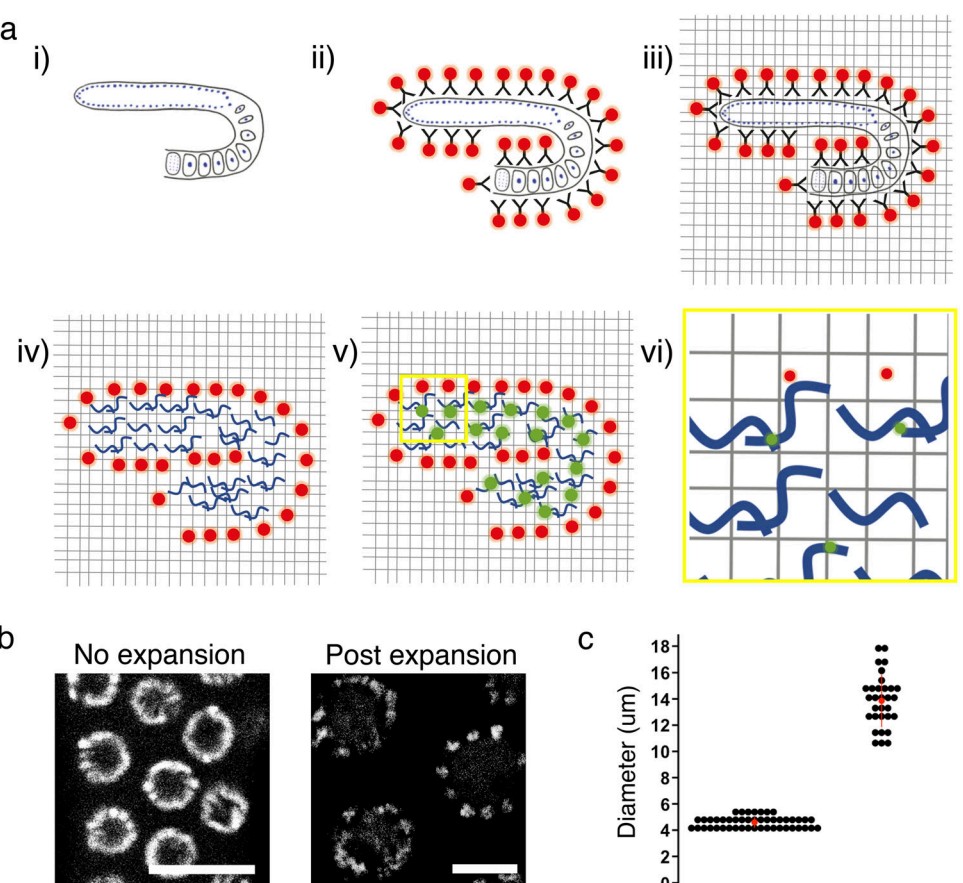

**Figure 1. Workflow of expansion microscopy on *C. elegans* germline.**
**(A)** Schematic of workflow. 3× EExM of immune-stained *C. elegans* germline with pan-protein stain. (i, ii) Dissected *C. elegans* germline tissues are stained with primary antibodies followed by fluorescently labelled secondary antibodies. (iii) Tissue is then chemically cross-linked to hydrogel which forms a mold of the tissue. (iv) Proteins within the tissue are digested before (v) pan-protein staining, whereby free amines are labelled with fluorescent dye. (vi) Hydrogel is expanded 3× and imaged. Resulting resolutions are xy ~40 nm and z ~100 nm. **(B)** DAPI staining of the pachytene nuclei in *C. elegans* germline pre- and post-expansion. Scale bar = 10 *μm*. **(C)** Nucleus diameter measured via DAPI staining to determine expansion factor. The average expansion factor across experiments is 3×. 31 expanded nuclei were measured from six independent experiments, and 47 non-expanded nuclei were measured from three independent experiments.
Source data are available for this figure.

Ouyang & Seydoux, 2022). Our pan-stain EExM pipeline is uniquely able to address this question because it allows us to observe perinuclear protein densities non-specifically. We examined proteins that are found in the three different granules: (1) PRG-1 which resides in the P granule, (2) MUT-16 (Mutator-16), an essential scaffold protein in the Mutator foci (Phillips et al, 2012), and (3) ZNFX-1 (zinc finger NFX1-type containing homolog), a member of the Z granule (Ishidate et al, 2018; Wan et al, 2018).

We first examined the localization of PRG-1 and MUT-16. We previously showed that PRG-1 and DEPS-1 condensates exist as small protein clusters that intertwine each other (Suen et al, 2020). As expected, PRG-1 is located in perinuclear protein densities (Fig 3A). We again observed that PRG-1 appears as small protein clusters, and these clusters in general distribute throughout the P granule, defined by the pan-protein staining.

piRNAs act as the initial recognition factor to identify mRNA transcripts. The effectiveness of the piRNAs is dependent on the secondary endo-siRNA pathway mediated by proteins found in the Mutator foci, MLOs with LLPS characteristics (Zhang et al, 2011; Phillips et al, 2012). MUT-16 is a scaffold protein shown to be essential in the formation of Mutator foci and that loss-of-function mutations in *mut-16* leads to a dramatic reduction in the level of the endo-siRNA 22Gs. It is well-established, via immunostaining for

Mutator foci and P granule protein markers, that Mutator foci and P granules are distinct granules. These granules are juxtaposed next to each other when examined using both conventional and super resolution fluorescent confocal microscopy (Phillips et al, 2012; Wan et al, 2018). We immunostained for MUT-16 as a marker for Mutator foci and PRG-1. In agreement with previous studies, MUT-16 appears as small foci and mostly do not overlap with PRG-1. Interestingly, MUT-16 foci are often located at the periphery of the germ granules (Fig 3A). To examine more closely as to whether MUT-16 is within the germ granule, we traced the boundary of the granule and overlaid it with the co-staining images of PRG-1 and MUT-16 (Fig 3B). We found that MUT-16 foci can lie both inside (granules 1, 3, and 4) and outside the germ granule boundary (granules 1 and 2). This is also reflected in the corresponding intensity plots where the MUT-16 signal peaks while the pan-protein stain decreases at the granule-cytosol boundary for granules 1 and 2.

Together with the Argonaute protein WAGO-4, ZNFX-1 plays an essential role in the transgenerational inheritance of RNAi (Ishidate et al, 2018; Wan et al, 2018). It was shown that ZNFX-1 segregates from the P granule protein PGL-1 at the Z2/Z3 embryonic stage, which leads to the formation of Z granules (Wan et al, 2018). We found that ZNFX-1 and PRG-1 exist within the same germ granule but occupy different areas, and some overlapping positions,

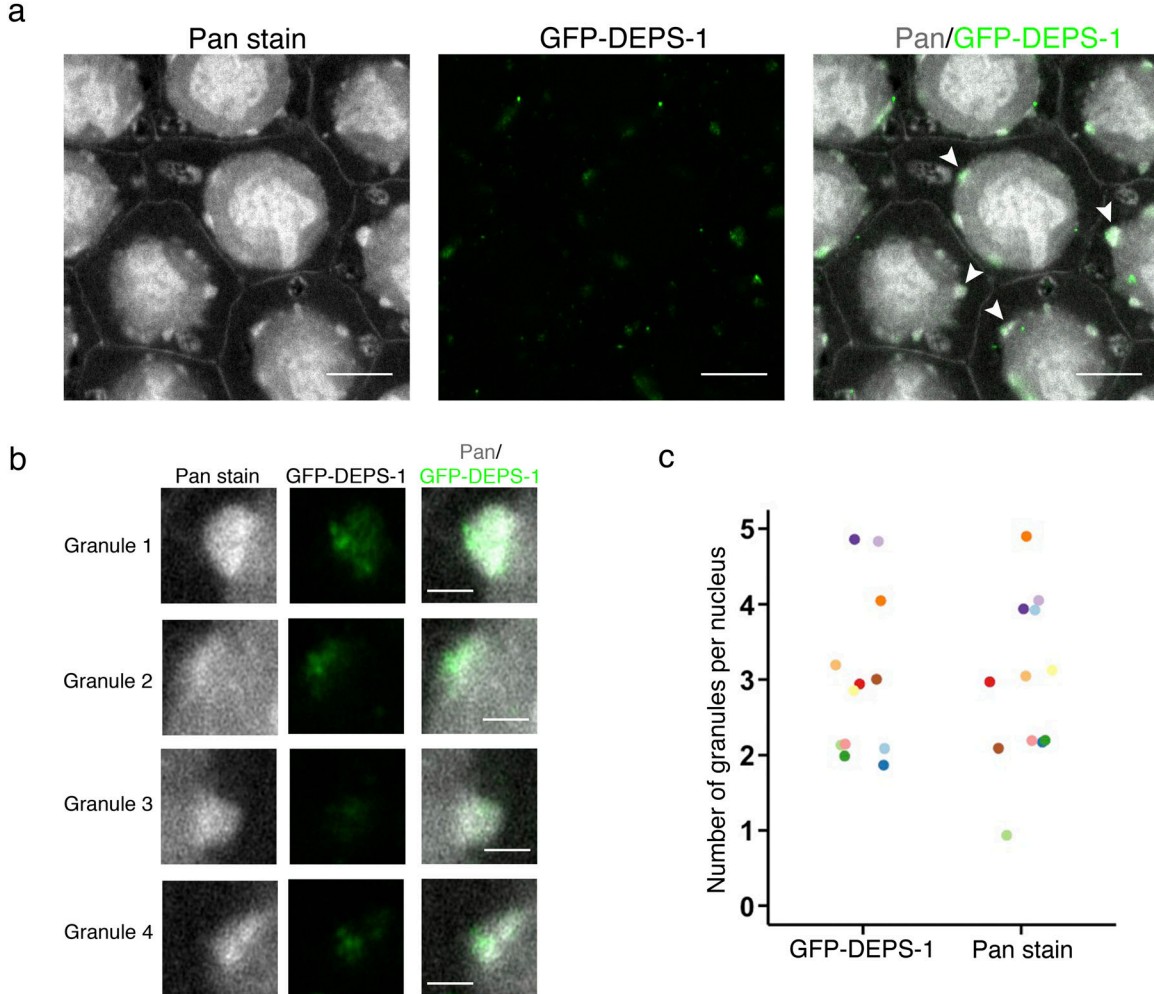

**Figure 2. Pan-protein staining reveals P granules as protein-dense perinuclear structures.**

**(A)** Pan-protein (NHS ester) and anti-GFP staining of animals expressing *gfp::deps-1*. Pan-protein staining reveals a number of features including P granules. GFP-DEPS-1 condensates (green) are localised to the P granule (gray). White arrows highlight P granules that are enlarged in (B). Scale bar = 10 *μ*m. **(B)** Zoomed images of P granules highlighted by arrowheads in (A). DEPS-1 condensates appear as small protein clusters that are localised to P granules. Scale bar = 2 *μ*m. **(C)** Number of granules observed per nucleus in a single optical slice. Granule is defined as perinuclear density observed either via GFP-DEPS-1 staining (green) or pan-protein staining (NHS ester; gray). 12 nuclei from four independent experiments were counted. Counts obtained from the same nucleus have the same colour.
Source data are available for this figure.

indicating that Z granules do not constitute separate protein dense structures in the pachytene region but in fact are compartments, or subdomains (Fig 4A; Ishidate et al, 2018) within the germ granule. Furthermore, ZNFX-1 frequently occupies the area closer to the cytoplasmic edge than the nuclear membrane edge of the P granule (Fig 4B). This is again reflected in the intensity plots in which ZNFX-1 can be seen to have a tendency localize toward the cytosolic end of the granule compared with PRG-1.

### Gross changes to germ granule morphology in *deps-1* and *mut-16* mutants

*deps-1* was first identified as a gene required for the correct localisation of PGL-1, which is a critical component of germ granules (Spike et al, 2008). How the germ granule as an organelle is affected by *deps-1* mutations is unknown. We performed pan-protein staining

on *deps-1 (bn124)* null animals to investigate whether the morphology of germ granules are affected (Fig 5A and B and Video 1 and Video 2). In *deps-1 (bn124)* null animals, perinuclear densities appear to be flattered overall compared with WT animals (Fig 5B and D). Furthermore, protein densities containing PRG-1 can be infrequently seen as granules almost dissociated from the nuclear membrane (Fig S2).

We previously showed that mutations in Mutator foci affect DEPS-1 perinuclear localization. Pan-protein staining reveals that similar to the *deps-1 (bn124)* mutant, the *mut-16 (pk710)* mutant animals also exhibit much diminished perinuclear protein densities (Fig 5C and D and Video 3). Given that *deps-1 (bn124)* mutation does not lead to a dramatic reduction in small RNA levels (Suen et al, 2020) but is required for the correct localization of other proteins (Spike et al, 2008; Suen et al, 2020), whereas *mut-16* mutation causes almost the complete loss of 22 Gs but not P granule protein localization, for example,

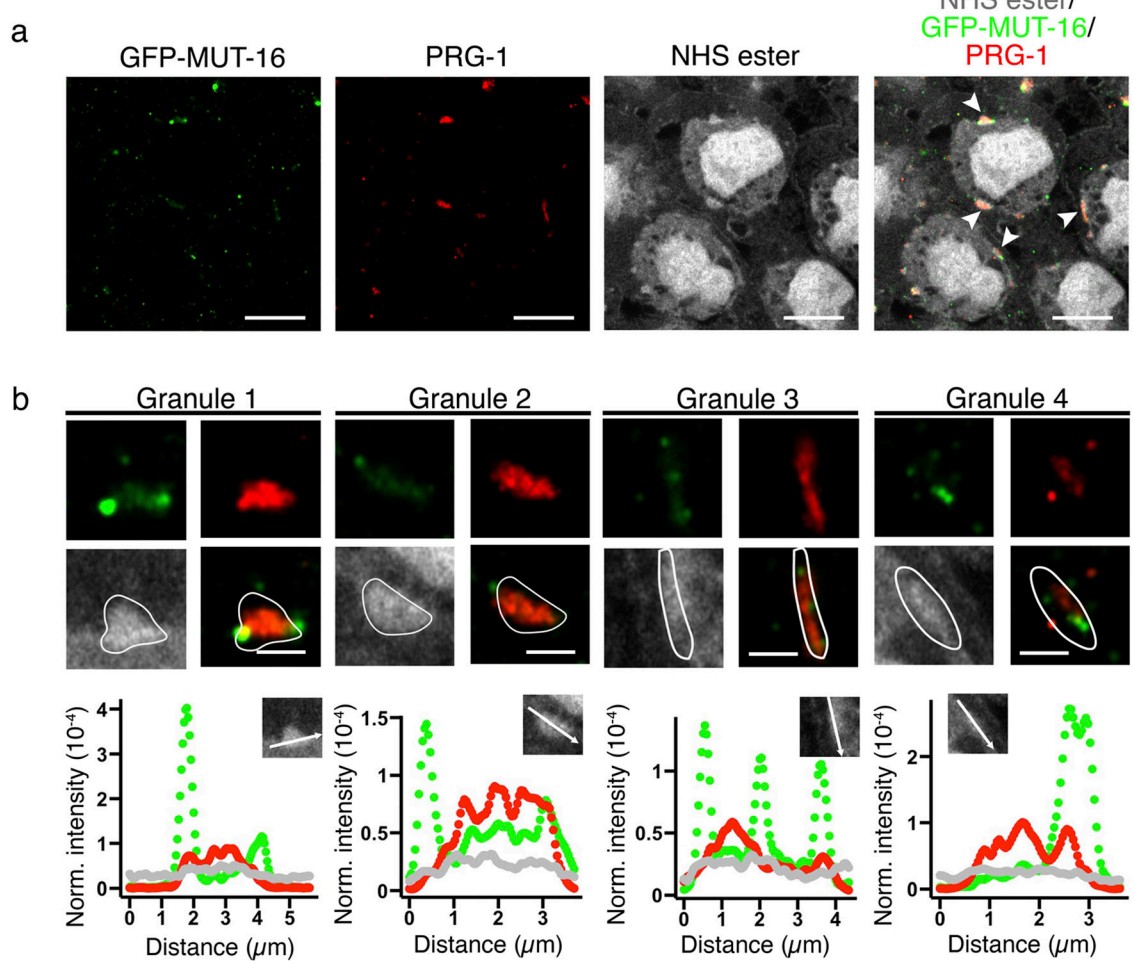

**Figure 3. Mutator foci are juxtaposed to PRG-1 condensates.**
**(A)** Pan-protein (NHS ester), anti–PRG-1, and anti-GFP staining of animals expressing GFP-tagged MUT-16. PRG-1 (red) and MUT-16 (green) colocalise to P granules (gray). MUT-16 and PRG-1 occupy distinct areas, whereby MUT-16 is frequently observed on the edge of the P granule space. White arrowheads in merged image highlight granules that are enlarged in (B). Scale bar = 10 μm. **(B)** Zoomed image of granules marked by arrowheads in (A). PRG-1 (red) exists as small cluster of proteins within the P granule. White line outlines the P granule boundary based on pan-protein staining (gray). MUT-16 (green) appears as single clusters that are either inside the P granule (granules 1, 3, and 4) or outside the P granule (granules 1 and 2). Scale bar = 2 μm. For each granule, the intensity of the staining was measured along the white arrow (inset of the plots) and normalized to the intensity of the entire granule to show the distribution of MUT-16 (green) and PRG-1 (red) relative to germ granule (gray). Source data are available for this figure.

PGL-1 and PRG-1 (Zhang et al, 2011; Phillips et al, 2012; Suen et al, 2020), the morphology of these perinuclear protein densities are dependent on both small RNA and protein levels.

## Discussion

The P granule in *C. elegans* has been fundamental for the discovery that MLOs can be formed by LLPS. It is a site in which the protein components of various small RNA pathways are located. Hence, understanding how small RNA-based epigenetics is facilitated or driven by LLPS in the germ granule will shed light on the biological relevance of LLPS.

Both electron and fluorescent light microscopies have played invaluable roles in the characterization of germ granules: EM led to the first observation of P granules as perinuclear densities that are

non-membrane bound (Schisa et al, 2001; Sheth et al, 2010), and fluorescent light microscopy has led to the identification of protein and RNA components in these perinuclear germ granules (Updike & Strome, 2010; Sundby et al, 2021). However, identifying the localization of specific proteins among ultrastructures revealed in EM is extremely challenging, and EM is limited in its accessibility preventing its wide spread use, for example, in screening the morphology of germ granules in mutants. Light microscopy, although it is accessible to non-specialist groups, relies on the immuno-staining for specific proteins as markers for specific compartments. In complex MLOs such as the germ granule, this "biased" approach does not allow examination of the organelle as a whole. In this work, we adapted pan-protein staining and EExM to overcome these limitations and present a workflow that can be easily applied to examine germ granules, among other cellular features, at nano-scale resolution.

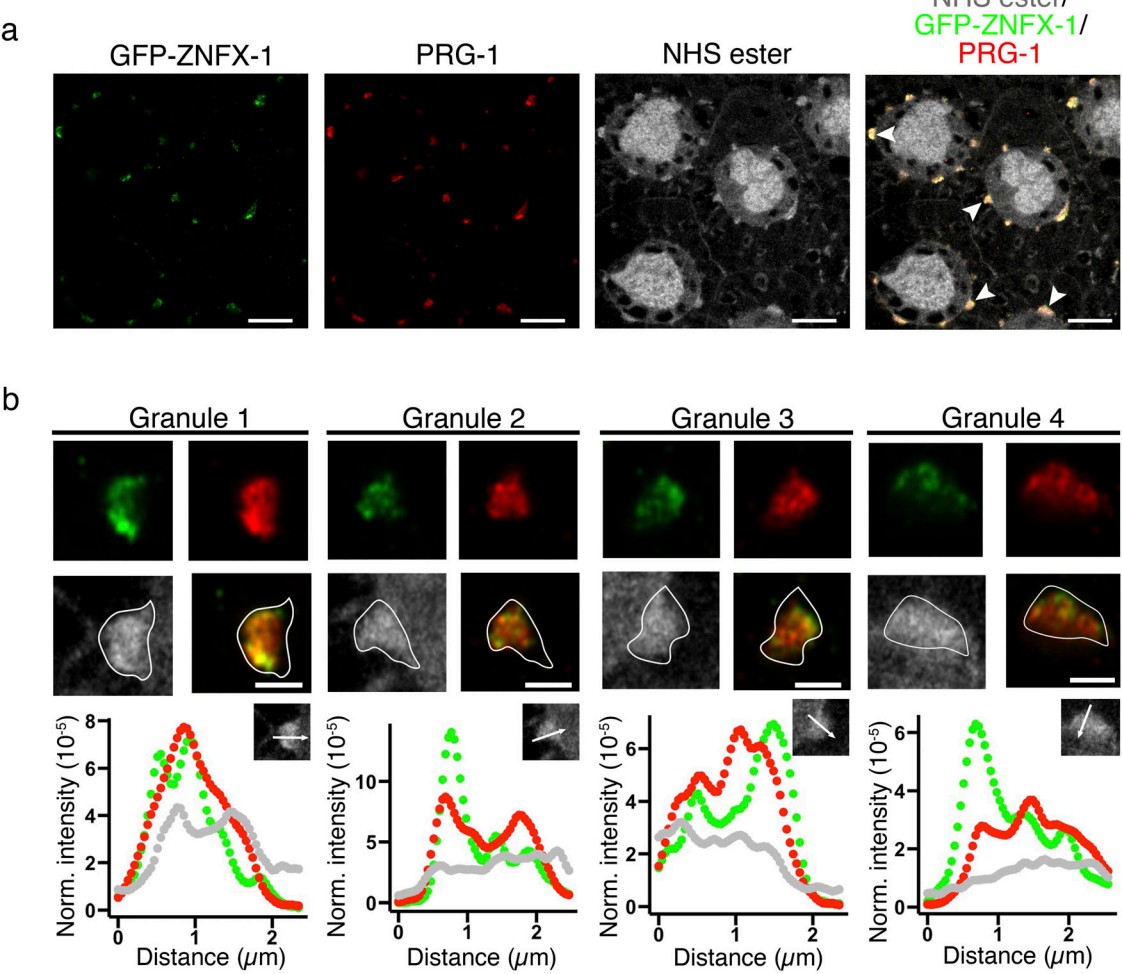

**Figure 4. ZNFX-1 and PRG-1 condensates are subdomains within the same germ granules.**
**(A)** Pan-protein (NHS ester), anti–PRG-1, and anti-GFP staining of animals expressing GFP-tagged ZNFX-1. PRG-1 (red) and ZNFX-1 (green) colocalise to P granules (gray). ZNFX-1 and PRG-1 occupy distinct, and overlapping areas within the P granule space. White arrowheads in merged image highlight granules that are enlarged in (B). Scale bar = 10 μm. **(B)** Zoomed image of granules marked by arrowheads in (A). Both PRG-1 (red) and ZNFX-1 (green) exist as small clusters of proteins within the P granule. The white line outlines the P granule boundary based on pan-protein staining (gray). ZNFX-1 is concentrated in areas closer to the cytoplasmic edge of the P granule than PRG-1 (granules 1, 3, and 4). Scale bar = 2 μm. For each granule, the intensity of the staining was measured along the white arrow (inset of the plots) and normalized to the intensity of the entire granule to show the distribution of ZNFX-1 (green) and PRG-1 (red) relative to germ granule (gray).
Source data are available for this figure.

We examined the localization of proteins that make up three small RNA pathways: PRG-1 for the piRNA pathway, MUT-16 for secondary endo-siRNAs, and ZNFX-1 for transgenerational inheritance of RNAi response. Whereas PRG-1 is known as a P granule protein due to its colocalization with PGL-1 (Batista et al, 2008), MUT-16 and ZNFX-1 are described as independent foci or granules that do not mix with PGL-1 (Phillips et al, 2012; Wan et al, 2018). Experimentally, whether or not Mutator foci or Z granule occupies the same germ granule is unclear. Here, we address this important question using our pan-protein staining with EExM workflow. We found that Mutator foci can exist both within and outside the perinuclear protein density marked by PRG-1 as P granule, whereas ZNFX-1 always exists within the same perinuclear protein dense structures as PRG-1. This means that the P and Z granules should be considered as subdomains of the same germ granules in the pachytene region of the germline, rather than independent granules. Furthermore, pan-protein staining provides context to protein localization by defining the border of the germ granule. We show that MUT-16 is often observed to be located at the periphery of the P granule close to the nuclear membrane. In contrast, ZNFX-1 can be found frequently concentrated to the cytoplasmic edge of the germ granule.

These observations raise two important questions. First, how should P granules, Z granules, and Mutator foci be defined? ZNFX-1 and MUT-16 clearly can occupy distinct areas from PRG-1, a P granule protein, in the perinuclear protein dense structure. However, conventionally, the P granule describes the entire perinuclear protein dense structure observed in EM studies (Schisa et al, 2001; Sheth et al, 2010). More recently, it is thought that distinct granules exist as condensates within the same germ granule (Ouyang &

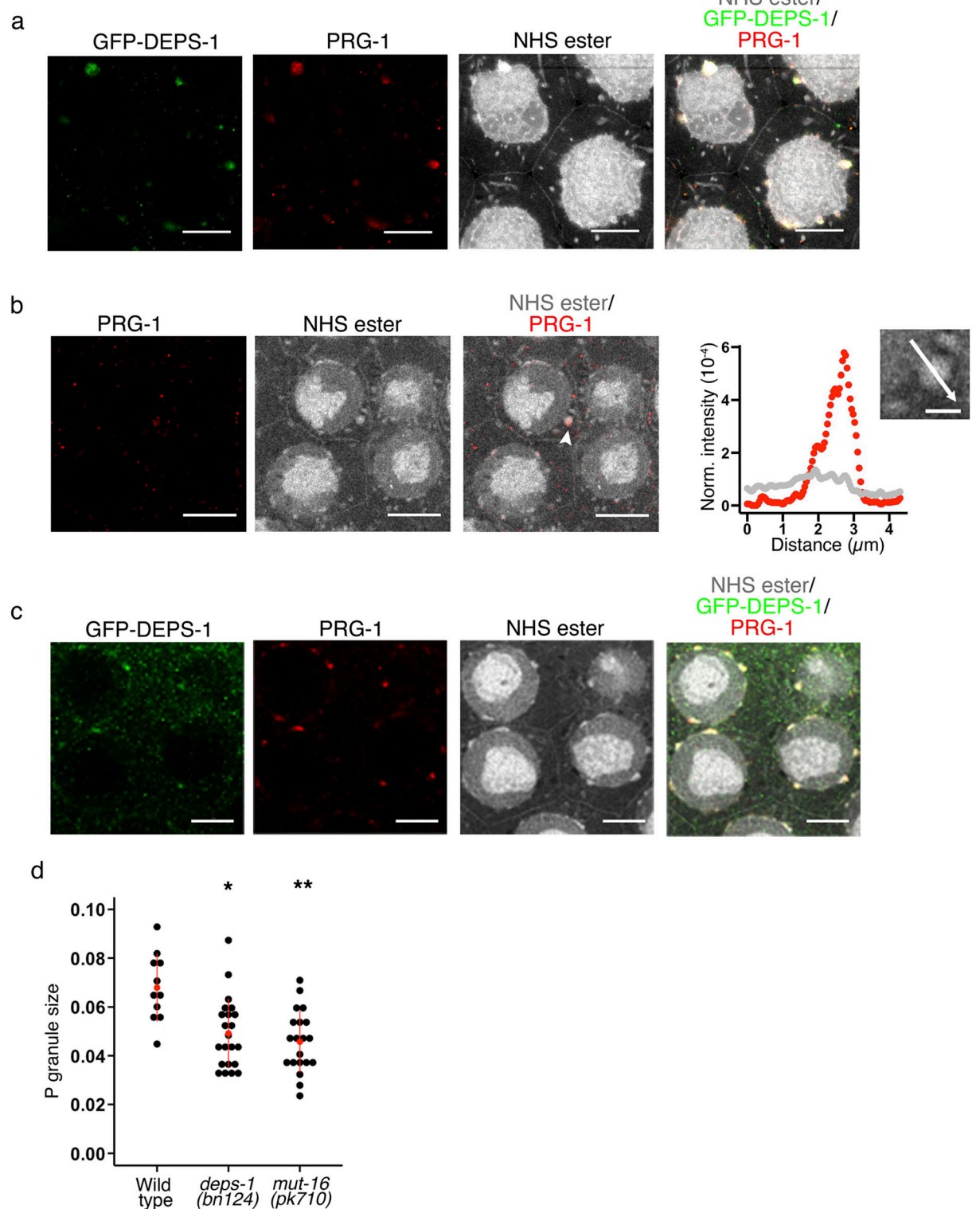

**Figure 5. P granules are malformed in animals defective in small RNA pathways.**
**(A)** Pan-protein (NHS ester) and anti-GFP staining of animals expressing *gfp::deps-1*. Scale bar = 10 *µm*. **(B)** Pan-protein (NHS ester) and anti–PRG-1 staining of *deps-1(bn124)* mutant animals. *deps-1* mutation leads to a reduction in the size of germ granules. Arrowheads highlight a germ granule that is mislocalised. Scale bar = 10 *µm*. The intensity of the staining was measured along the white arrow (inset of the plot; scale bar = 2 *µm*) and normalized to the intensity of the entire granule to show the distribution of PRG-1 relative to the germ granule. **(C)** Pan-protein (NHS ester), anti-DEPS-1, and anti–PRG-1 staining of *mut-16 (pk710)* mutant animals. *mut-16* mutation leads to a reduction in the size of germ granules. Scale bar = 10 *µm*. **(D)** P granule size in WT animals, *deps-1 (bn124)*, and *mut-16 (pk710)* mutants. P granule size was

Seydoux, 2022), a concept that our findings here support at least in the pachytene region of the germline. Many proteins have been classified as "P granule" proteins primarily using confocal microscopy with PGL-1 as a protein marker for P granules. With the advent of super resolution microscopy, it would be worth re-examining the localization of these P granule proteins. This might lead to the identification of other sub-domains/condensates within the P granule.

Second, because these granules co-exist in the same germ granule, it remains to be determined how Mutator foci, Z granule, and P granule proteins segregate into different areas. Feric et al showed that the compartments within the nucleolus are linked to the steps in ribosomal biogenesis, and this segregation is driven by the differences in droplet surface tensions in vitro (Feric et al, 2016). Given that RNA species, both long and short, are main players in the epigenetic pathways, it would be reasonable to hypothesize that RNA contributes to the dynamic formation of subdomains within the P granule.

The contribution of RNA in P granule structural integrity is supported by the gross morphological changes observed in *mut-16* mutant animals, which exhibits a significant loss in specific small RNA populations (Zhang et al, 2011). Phillips et al previously showed that PGL-1 localisation is not disrupted by mutations in Mutator foci genes, including *mut-16* (Phillips et al, 2012), suggesting that the reduced size of P granules might be driven by the loss of small RNA. In contrast, the *deps-1* mutation leads to a loss of PGL-1 from the perinuclear region (Spike et al, 2008) while exhibiting a less significant loss in small RNAs compared with *mut-16* mutants (Suen et al, 2020). Hence, morphology of the germ granule is dependent on both proteins and RNA.

Although previous EM studies show that the P granule consists of a crest and a base (Sheth et al, 2010), we were not able to observe these features using pan-protein staining. It will be interesting to examine the P granule at a higher resolution through higher orders of EExM, which might allow us to observe these features; however, the risks of spatial distortions that accompany the combination of multicellular tissue imaging with 10× or greater EExM need to be carefully managed (Sheard et al, 2021 Preprint). It is also possible that the crest is composed of other materials, for example, RNA, which would not be stained by the pan-protein stain but could be labelled by nucleic acid–based approach.

Our study has focused on observing subdomains within the P granule and determining the effect of small RNA pathway mutants. However, a number of cellular features are visible under pan-protein staining, and it would be important to identify these. Furthermore, although we showed using DAPI staining that expansion of nuclear compartment is isotropic in the x- and y-planes, we neither carried out isotropic assessment for other features observed using pan-protein staining nor for the z-plane. It would be essential to measure expansion isotropy for the specific cellular compartments of interest in future studies (Büttner et al, 2021).

# Materials and Methods

## General animal maintenance

Animals were fed with HB101 and maintained at 20°C on NGM plates. Strains used in this study are listed in Table S1.

## Worm dissection and immunostaining

Microscope slides were incubated with 0.01% poly-lysine solution for 1 h at room temperature, and excess liquid was removed by tissues. 1-d-old adults were dissected for germline on the poly-lysine coated slides in 9 $\mu$l of 1 mM levamisole (diluted in M9) using 21G needles to remove their heads or tip of the tails. Extruded germlines were then fixed with 4% paraformaldehyde for 10 min at room temperature. After 10 min, the solution was diluted three-fold, using a gel-loading pipette tip to remove excess liquid (i.e. leaving 9 $\mu$l). A rectangular coverslip was then put on top of the dissected samples perpendicular to the slide and placed on top of a pre-chilled metal block (chilled by dry ice) for at least 10 min. The slides were then removed from the metal block and coverslip lifted up in one smooth motion. The slides were placed in prechilled methanol at −20°C for 20 min. Fixed samples were washed with PBS-T (PBS supplemented with 0.1% tween-20) before primary antibody addition. Primary antibodies were incubated with the samples at 4°C for overnight. Secondary antibodies and DAPI were incubated at 37°C for 1 h in the dark. Antibodies used were as folllows: anti–PRG-1 (1:500; Custom) and anti-mouse GFP (A-11120; 1:500; Thermo Fisher Scientific). Anti-rabbit Alexa Fluor 594 secondary antibody (A11012; Thermo Fisher Scientific) was used at 1:500, and anti-mouse Atto 647 secondary antibody (50185; Sigma-Aldrich) was used at 1:160.

## Gel expansion

Dissected, immunostained germlines mounted on microscope slides were washed twice PBST and then twice with PBS. The germlines were incubated with 0.1 mg/ml Acryloyl-X SE (stocks stored as 10 mg/ml in DMSO and diluted to working concentration in PBS; A20770; Thermo Fisher Scientific) overnight at 4°C. Slides were washed three times with PBS next afternoon. Excess liquid was removed as much as possible without drying the germlines before the addition of monomer solution (8.6 mg/ml sodium acrylate (461652; Fluorochem), 2.5 mg/ml acrylamide, 0.15 mg/ml N,N'-methylenebisacrylamide (M7279; Merck), and 11.7 mg/ml NaCl in 1× PBS). Samples were incubated in monomer solution for 30 min at 4°C. Monomer solution was then removed as much as possible. 100 $\mu$l of polymer solution (94 $\mu$l monomer solution, 4 $\mu$l 1× PBS, 1 $\mu$l 10% APS, and 1 $\mu$l 10% TEMED) was added to the sample and a coverslip was gently placed on top of the sample. Gelation was performed in a humid chamber at 37°C for 2 h. The coverslip was carefully removed using a razor blade. The gel was detached from the slide

---

calculated by measuring the maximum length of the P granule perpendicular to the nuclear membrane and normalizing it to the diameter of the nucleus. P granules are smaller in both mutants compared with WT animals. *P-value < 0.001 and **P-value < 0.0001.
Source data are available for this figure.

using a razor blade and placed into a small Petri dish. Digestion buffer (50 mM Tris, pH 8.0, 1 mM EDTA, 0.5% Triton X-100, 0.8 M guanidine HCl, and 1:100 proteinase K) was added in sufficient amount to cover the gel to digest the germline and the gel was digested overnight at room temperature. Digestion solution was removed the next morning, and the gel was washed with PBS twice, transferred to a new Petri dish, and washed with 100 mM sodium bicarbonate twice. The gel was then incubated in 10 $\mu$M NHS ester Atto 488 in 100 mM sodium bicarbonate with gentle rocking at room temperature for 1.5 h. The gel was washed with PBS once and transferred to a larger Petri dish (e.g. 10 cm petri dish). Deionised water was added to the dish three–five times, 30–60 min each until expansion reaches plateau.

### Confocal microscopy

50-mm $\mu$-dishes (81131; Ibidi) were coated with 0.1% poly-lysine to prevent drift during imaging. Images were taken on a Zeiss LSM880 inverted microscope using the Airyscan mode using either a 40× or 60× objective. Lasers used were HeNe 633 for the 647 channel, DPSS 561 for the 594 channel, and argon for 488 channel.

### Germ granule size analysis

The maximum height of the P granule perpendicular to the nuclear membrane was measured. To account for the expansion factor in the individual samples, the height of the P granule is divided by the diameter of the nucleus.

### Germ granule protein distribution analysis

Using ImageJ, a line was drawn across a germ granule. The intensity of protein signals was measured across the line using the Plot Profile plugin in ImageJ. The integrated signal of the entire granule of interest was also measured in ImageJ. To calculate the distribution of the protein of interest, intensity along the line was expressed as a fraction of the integrated intensity of the granule.

### Expansion isotropy analysis

Automatic threshold was applied to pre- and post-expansion images of nuclei stained with DAPI using ImageJ. Aspect ratio of the threshold images of the nuclei was then measured using the Analyze particle function.

## Data Availability

Original imaging data can be found in Mendeley data, https://doi.org/10.17632/3djrv7v4dj.1.

## Supplementary Information

## Acknowledgements

We would like to thank Ruth Huges and Sally Boxall at the Bio-Imaging Facility for technical support. JE Ladbury was supported by Cancer Research UK (Grant C57233/A22356). I Jayasinghe was supported UKRI (Grant: MR/S03241X/1); I Jayasinghe, TMD Sheard, and KM Suen were supported by the Integrated Biological Imaging Network (IBIN5IJ).

### Author Contributions

KM Suen: conceptualization, data curation, formal analysis, funding acquisition, validation, investigation, methodology, and writing—original draft, review, and editing.
TMD Sheard: conceptualization, formal analysis, funding acquisition, validation, and investigation.
C-C Lin: formal analysis and investigation.
D Milonaityte: formal analysis and investigation.
I Jayasinghe: conceptualization, data curation, formal analysis, supervision, funding acquisition, investigation, and writing—original draft, review, and editing.
JE Ladbury: supervision, investigation, project administration, and writing—original draft, review, and editing.

### Conflict of Interest Statement

The authors declare that they have no conflict of interest.

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
