## [Reviewer comments · Life Science Alliance]

Life Science Alliance

Expansion microscopy reveals subdomains in *C. elegans* germ granules

Kin Man Suen, Thomas Sheard, Chi-Chuan Lin, Dovilė Milonaitytė, Izzy Jayasinghe, and John Ladbury

DOI: <https://doi.org/10.26508/lsa.202201650>

Corresponding author(s): John Ladbury, University of Leeds and Kin Man Suen, University of Leeds

Review Timeline:

Submission Date:	2022-08-04
Editorial Decision:	2022-09-12
Revision Received:	2022-12-21
Editorial Decision:	2023-01-19
Revision Received:	2023-01-25
Accepted:	2023-01-26

Scientific Editor: Novella Guidi

Transaction Report:

September 12, 2022

Re: Life Science Alliance manuscript #LSA-2022-01650-T

Prof. John E Ladbury
University of Leeds
Cellular & Molecular Biology
Room 7.08, Miall Building
Leeds, North Yorkshire LS2 9JT
United Kingdom

Dear Dr. Ladbury,

Thank you for submitting your manuscript entitled "Expansion microscopy reveals subdomains in *C. elegans* germ granules" to Life Science Alliance. The manuscript was assessed by expert reviewers, whose comments are appended to this letter. We invite you to submit a revised manuscript addressing the Reviewer comments.

Thank you for this interesting contribution to Life Science Alliance. We are looking forward to receiving your revised manuscript.

Sincerely,

B. MANUSCRIPT ORGANIZATION AND FORMATTING:

Reviewer #1 (Comments to the Authors (Required)):

This submission by Kin Suen et al., entitled "Expansion microscopy reveals subdomains in *C. elegans* germ granules," introduces a new technique to image the *C. elegans* germline. Here, a version of super-resolution expansion microscopy is applied that combines the specificity of immunostaining with the ability to observe protein density (proExM). This new technique is applied to correlate germ granule descriptions with a spatial map of individual germ granule proteins. In doing so, the authors can address the nature of germ-granule sub-compartments that several labs have recently described. This technical advance is substantial and will likely be employed by numerous laboratories studying the *C. elegans* germline and germ granules. While enthusiasm is very high for this work, there are significant concerns to address. The first is a matter of semantics brought on by this newfound ability to correlate protein density with sub-granule distribution. Until now, EM and immunostaining (apart from limited CLEM techniques) were separate approaches to looking at germ granule and sub-granule localization. The term 'P granule' has traditionally been used to describe the electron-dense perinuclear regions of germ cells. With the discovery of Mutator foci, Z granules, SIMR granules, and other sub-domains, the term 'P granule' has become more correlative of specific germ-granule protein components like PGLs, GLHs, DEPS-1, PRG-1; Z granules ZNFX-1 and WAGO-4; MUT-16 foci; SIMR-1. The authors referring to their protein-dense perinuclear granules and P granules introduce unnecessary confusion. One solution the authors should consider to add clarity to their findings is to use the term 'germ granule,' which is in the title of their manuscript, to describe the germline-specific, protein-dense granules revealed by this new proExM technique. 'Germ granule' is used more frequently in this field to describe the P/Z/M/S sub-granule collective in the adult *C. elegans* germline. The authors should seize the opportunity to clarify the use of terms in a way that can be adopted by their field. Another major criticism is the descriptive nature of the paper-statements throughout the paper needed to be backed up by quantifiable data and image analysis pipelines. The data points need to be substantially increased when images are quantified in Figures 2C and 4D. The number, size, and characteristics of germ granules would be much more thorough using volume data from z-stacks rather than single-plane images. And finally, as this work introduces new practical techniques, image acquisition parameters should be extensive, including Airyscan parameters, objective information, filter sets, z-step size, exposure times, and more.

Specific points:

- 1) A minor curiosity that might be worth discussing is whether the expansion step is necessary for dissected germlines - that is, does pan-protein staining work following the protease step even if it isn't followed up by swelling? With super-resolution techniques, the added 3x resolution may not wholly offset potential artifacts introduced by expansion. This wasn't addressed in the first paragraph of the results, where the limitations on whole-mount worms were discussed.
- 2) In paragraph two of the results, what information is used to derive the ~40nm resolution? Is this experimentally determined, or was it derived by dividing the Zeiss published Airyscan resolution of ~120nm by the 3-fold expansion?
- 3) "DEPS-1 is a scaffold protein essential for P granule formation (Spike 2008)" is an incorrect statement. This paper showed that levels of GLH-1 are severely depleted and that PGL-1 granule size is severely reduced, but as evidenced in their figures of the adult germline, GLH-1 and PGL-1 granules are still present in the adult germline of *deps-1* animals.
- 4) Germ granule boundaries traced in figure 3 are somewhat arbitrary and would benefit from automated thresholding. For example, statements describing granule proximity to a defined boundary should be determined, quantified, and supported using automated image analysis pipelines and increased data points. This would also apply in Figure 4 where granule flattening and dissociation from the nuclear periphery are discussed.
- 5) In the second to the last paragraph of the results, the DEPS-1 statements on any presumable P-granule assembly pathway are more directly related to the early embryo. In the adult germline, P granule assembly is more synergistic than linear, and the hierarchy is less clear. For example, DEPS-1 can be mislocalized when other P granule proteins are depleted.
- 6) In the discussion of nucleolar compartments, it may be more accurate to state, "Feric et al. shows that the compartments within the nucleolus are linked to the steps in ribosomal biogenesis and this segregation is driven, at least in vitro, by the differences..."
- 7) Figure 4 and Sup fig 1 should include wild-type images for better comparison and labels on the figure to indicate wild-type vs. mutant images. Also, improved quant as discussed above.

Referee Cross-Comments

I concur with the points from revs 2 & 3.

Reviewer #2 (Comments to the Authors (Required)):

Suen et al. utilized previously established techniques for expansion microscopy to visualize perinuclear nuage in dissected *C. elegans* germlines. Using a pan-proteome stain to define "P granules" and protein-specific fluorescent markers and antibodies, the authors investigate the distribution of various nuage proteins. They conclude that some markers define sub-domains of P granules while others (MUT-16) reside outside or at the edge. These observations are intriguing but not conclusive, especially since the terms used (P granule, compartment, nuage) are poorly defined. Also it is not clear what these observations add to the literature given that it was already known that these markers do not co-localize in nuage, and that nuage is a multi-condensate structure. We recommend the following:

- Define terms. It is not clear that the proteome stain corresponds to "P granules", which traditionally have been defined by PGL-1 staining. The EM studies in the past did not distinguish between P/Z granules and mutator foci and it is likely that the pan-proteome stain does not either. It may be more prudent to use the more generic term nuage to refer to the proteome staining, and use the protein-specific reagents to define condensates within the nuage, as is now the norm (e.g. Ouyang and Seydoux, RNA 2022). Different granules/condensates can have distinct compositions (PGL-1 vs ZNFX-1) yet likely have the same protein density.
- Quantify results to justify conclusions. How often are MUT-16 inside vs at the edge of the pan-proteome stain?
- Provide control Airy-scan images of non-expanded samples to 1) demonstrate isotropic expansion in all compartments and axes, and 2) illustrate the advantage of the expansion technique. Expansion factors can differ for different organelles within the same cell (Büttner et al., 2020). The authors should show what controls were done to ensure the gel expansion was isotropic. This is especially important since nucleus diameter was used to normalize the granule size in Fig. 4.
- The authors find that there is an average of 2.5 granules per nucleus in a Z-slice, however this data is not well represented by their images. Most nuclei shown appear to contain more than 2 granules. Additionally, previously published reports of non-expanded germline nuclei also appear to contain more than 2 granules per nucleus in a Z-slice. The authors should comment on why expanded nuclei might appear to have fewer granules than non-expanded.
- Fig. 3B: NHS ester staining does not appear to clearly define the boundary of the P granule. As seen in several examples throughout the paper (especially granule 3 and 4 in Fig. 3B), the edges of the perinuclear densities are blurred together with the nuclei. Before coming to conclusions that mutator foci are within P granules, the boundary of the P granule should be better defined.
- Fig. 3C, 3D: Toraason et al., 2021 found that the frequency of ZNFX-1 colocalization with P granules changes depending on relative gonad position. The authors should comment on what region of the gonad they imaged these granules and how often they observed colocalization vs. non-colocalization.
- Fig. 4: The authors should include micrographs of wild-type granules as a comparison for these mutants. Based on the presented images, the mut-16 mutant granules do not appear smaller than wild-type or depts-1 mutants, even though that's what is shown in the quantification.
- Fig. 4D: The units for measuring P granule size is not given. If in microns, can such small differences (less than 40 nm) in granule size be accurately measured given the authors stated they have 40 nm resolution in the xy plane? Also, the authors should show the same experiment in non-expanded germlines to illustrate the necessity of using expansion microscopy to measure these differences.
- In the text (last paragraph, pg. 7) the authors state that mutations in mutator foci affect DEPS-1 localization. Later in the paragraph they say that mut-16 mutation causes loss of 22Gs but not P granule protein localization. These statements seem contradictory. Additionally, there is not enough evidence given to show that P granule morphology is dependent on small RNA levels, since loss of mut-16 also affects DEPS-1 localization (Suen et al., 2020).
- Fig. 4B: Based on the NHS ester staining, this P granule does not appear disconnected from the nucleus.

Reviewer #3 (Comments to the Authors (Required)):

Summary:

Suen et al. present the identification of subdomains in adult *C. elegans* germ granules using an advanced expansion microscopy coupled with bulk labeling of the proteome. Based on colocalization study of the Z granule protein, P granule and Mutator foci protein markers, these subdomains are localized within the same perinuclear protein-dense structures. The size and morphology of these perinuclear protein-dense structures are dependent on both small RNA and protein levels. The enhanced expansion microscopy pipeline developed in this study provides an important tool and a significant advance in further understanding of biogenesis, spatiotemporal regulation and function of the membraneless organelles and liquid-like biomolecular

condensates.

There are some substantial concerns and minor comments about figures, interpretation and clarity of writing that would need to be addressed.

Essential revisions:

To the best of my knowledge, this is the first reported implementation of the expansion microscopy workflow to visualize germ granules; therefore validation of isotropy and benchmarking of the overall distortion associated with the entire protocol is required. The P granule diameter distribution after expansion is not linearly preserved (Fig. 1c), this could potentially distort the relative organization of proteins and relative length measurements. Would it be possible to normalize after-expansion feature measurements and to estimate the error of measurements?

Minor comments:

1. Fig. 1b and Fig.1c need to be referred in the Results section.
2. Would it be possible to add % of the surface overlap between NHS ester staining and GFP-DEPS-1 in P granules (Fig. 2)? Colocalization of GFP-DEPS-1 and PRG-1 in the wild-type animals needs to be shown (Fig. 2).
3. The P granule morphology appears variable in the GFP-reporter strains (GFP-DEPS-1 Fig. 2a vs. GFP-MUT-16 and GFP-ZNZF-1 in Fig. 3a-b). Does this reflect expected background level of the P granule heterogeneity?
4. The P granule boundaries are defined by the local protein density (Fig 3b). Is the colocalization pattern of PRG-1 with GFP-MUT-16 and GFP-ZNZF-1 consistent with DEPS-1 staining?
5. The solid lines outlining P granules (Fig. 3b and 3d) need to be replaced by thin dashed-lines that offer better visibility and ease evaluation of the partially colocalized features.
6. The authors describe the phenotypic consequence of loss of DEPS-1 and MUT-16 (Fig. 4). It would be informative to have quantification of the average granule number/retention per nucleus in wild-type animals, *deps-1* and *mut-16* mutants included in Fig 4.

Referee Cross-Comments:

Overall, there is a substantial overlap and a very balanced complementarity between reviewers' reports. Both, reviewer #1 and reviewer #2 raised important concerns and questions regarding validation of the new technique used in this study, adult *C. elegans* germ granule biology, accuracy and precision in the terminology, the lack of robust quantitative image analysis and data interpretations.

I strongly agree with their comments, that need to be addressed by the authors; nothing to flag specifically.

Reviewer #1 (Comments to the Authors (Required)):

This submission by Kin Suen et al., entitled "Expansion microscopy reveals subdomains in *C. elegans* germ granules," introduces a new technique to image the *C. elegans* germline. Here, a version of super-resolution expansion microscopy is applied that combines the specificity of immunostaining with the ability to observe protein density (proExM). This new technique is applied to correlate germ granule descriptions with a spatial map of individual germ granule proteins. In doing so, the authors can address the nature of germ-granule sub-compartments that several labs have recently described. This technical advance is substantial and will likely be employed by numerous laboratories studying the *C. elegans* germline and germ granules.

While enthusiasm is very high for this work, there are significant concerns to address. The first is a matter of semantics brought on by this newfound ability to correlate protein density with sub-granule distribution. Until now, EM and immunostaining (apart from limited CLEM techniques) were separate approaches to looking at germ granule and sub-granule localization. The term 'P granule' has traditionally been used to describe the electron-dense perinuclear regions of germ cells. With the discovery of Mutator foci, Z granules, SIMR granules, and other sub-domains, the term 'P granule' has become more correlative of specific germ-granule protein components like PGLs, GLHs, DEPS-1, PRG-1; Z granules ZNFX-1 and WAGO-4; MUT-16 foci; SIMR-1. The authors referring to their protein-dense perinuclear granules and P granules introduce unnecessary confusion.

One solution the authors should consider to add clarity to their findings is to use the term 'germ granule,' which is in the title of their manuscript, to describe the germline-specific, protein-dense granules revealed by this new proExM technique. 'Germ granule' is used more frequently in this field to describe the P/Z/M/S sub-granule collective in the adult *C. elegans* germline. The authors should seize the opportunity to clarify the use of terms in a way that can be adopted by their field.

We have taken Reviewer #1 and #2's very helpful suggestions to clarify our language. We have replaced 'P granule' and 'perinuclear protein density' with 'germ granule'. We kept the term P granule in certain instances only to help general readers i.e. those not from the *C. elegans* community, when specific references are made for past studies on P granules. We have also kept the term 'perinuclear protein density' in a few places in the first section of the Results. This is because we were determining whether those perinuclear density were in fact germ granules in that section. Furthermore, we have cited Ouyang & Seydoux 2022, a highly relevant review, throughout our manuscript.

Another major criticism is the descriptive nature of the paper-statements throughout the paper needed to be backed up by quantifiable data and image analysis pipelines. The data points need to be substantially increased when images are quantified in Figures 2C and 4D.

We have added 4 data points from two different germlines in Fig 2C. We wish to point out the original quantification was obtained from 3-4 independent dissections, which is typical of confocal microscopy experiments in the field.

In the original Fig 4D (which is now Fig 5C), the number of granules quantified in this study is more than sufficient for power of 80% with type 1 error set at 0.05.

The number, size, and characteristics of germ granules would be much more thorough using volume data from z-stacks rather than single-plane images. And finally, as this work introduces new practical techniques, image acquisition parameters should be extensive, including Airyscan parameters, objective information, filter sets, z-step size, exposure times, and more.

In principle, volume data would be more impactful and it was our original intention to record and analyse our data using z-stacked data. However, in expansion microscopy, the technicality of imaging z-stacked images that can capture entire granules consistently is extremely challenging because of the thickness of the germline tissue (which is now 3x thicker to image). The thickness of the sample means that images have to be taken over long periods of time during which bleaching occurs and the hydrogel is prone to movement (imaging faster would result in loss of resolution). Having tried it for months, we felt it was more prudent and indeed sufficient to make comparisons using single-sliced data to achieve our goal of introducing this technique to the wider worm germ granule community, which includes many groups that excel at microscopy and can no doubt build on our work.

We have included videos of z-stacked images of wild-type, *mut-16* and *deps-1* mutants.

We have also included general information on the objective and laser used in Materials and Methods. Images will also be deposited to Mendeley Data as .czi files. This will allow the readers to download the metadata files which contain specific settings for each of the images (or even reload our settings on a Zeiss instrument). This would be a more efficient way of sharing the settings used. Finally, we would like to stress that the novelty of our work does not stem from image acquisition, but rather the methodology of carrying out expansion microscopy on dissected germline for which we have written a detailed explanation in Materials & Methods.

Specific points:

1) A minor curiosity that might be worth discussing is whether the expansion step is necessary for dissected germlines - that is, does pan-protein staining work following the protease step even if it isn't followed up by swelling? With super-resolution techniques, the added 3x resolution may not wholly offset potential artefacts introduced by expansion. This wasn't addressed in the first paragraph of the results, where the limitations on whole-mount worms were discussed.

No, the germ granules are not easily identifiable without expansion (Fig. S1).

2) In paragraph two of the results, what information is used to derive the ~40nm resolution? Is this experimentally determined, or was it derived by dividing the Zeiss published Airyscan resolution of ~120nm by the 3-fold expansion?

It is derived from Zeiss. We have removed this from the results and introduction.

3) "DEPS-1 is a scaffold protein essential for P granule formation (Spike 2008)" is an incorrect statement. This paper showed that levels of GLH-1 are severely depleted and that PGL-1 granule size is severely reduced, but as evidenced in their figures of the adult germline, GLH-1 and PGL-1 granules are still present in the adult germline of *deps-1* animals. In Spike et al. 2008 figures 1 B and D, in the pachytene region there are significantly fewer PGL-1-containing granules in the perinuclear region with more PGL-1 diffused staining in the

cytoplasm. This is the same for GLH-1 staining in figures 4 B and D. Furthermore, *deps-1* animals exhibit germline-related phenotypes that are associated with the germ granules. Perhaps there are some PGL-1/GLH-1 containing granules can be observed and they are likely to be abnormal in their content given the phenotypes.

However, we have changed 'essential' to 'important' and 'formation' to 'assembly', which is a term used in the title of this study.

4) Germ granule boundaries traced in figure 3 are somewhat arbitrary and would benefit from automated thresholding. For example, statements describing granule proximity to a defined boundary should be determined, quantified, and supported using automated image analysis pipelines and increased data points. This would also apply in Figure 4 where granule flattening and dissociation from the nuclear periphery are discussed.

We have now included analysis for the localisation of PRG-1, ZNFX-1 and MUT-16 (now figures 3 and 4.). This was carried out by using the 'plot profile' function in Image J to measure the intensity along a line that runs across the signals from protein staining. Intensity was then normalised to the signals from the whole granule, i.e. the plots reflect the distribution of the specific proteins (this is to account for the difference in staining from different antibodies/dyes). The same was carried out for granule dissociation (originally Fig 4, now Fig 5 and S2). However, to measure the flattening of granules using an automatic pipeline is out of the scope of this manuscript.

5) In the second to the last paragraph of the results, the DEPS-1 statements on any presumable P-granule assembly pathway are more directly related to the early embryo. In the adult germline, P granule assembly is more synergistic than linear, and the hierarchy is less clear. For example, DEPS-1 can be mislocalized when other P granule proteins are depleted.

We have adjusted the text in accordance with the Reviewer's comments.

6) In the discussion of nucleolar compartments, it may be more accurate to state, "Feric et al. shows that the compartments within the nucleolus are linked to the steps in ribosomal biogenesis and this segregation is driven, at least in vitro, by the differences..."

We have adjusted the text in accordance with the Reviewer's comments.

7) Figure 4 and Sup fig 1 should include wild-type images for better comparison and labels on the figure to indicate wild-type vs. mutant images. Also, improved quant as discussed above.

We thank the Reviewer for pointing out the error in not including the wild-type images. It is included in Fig 5a.

Referee Cross-Comments

I concur with the points from revs 2 & 3.

Reviewer #2 (Comments to the Authors (Required)):

Suen et al. utilized previously established techniques for expansion microscopy to visualize perinuclear nuage in dissected *C. elegans* germlines. Using a pan-proteome stain to define "P granules" and protein-specific fluorescent markers and antibodies, the authors investigate the distribution of various nuage proteins. They conclude that some markers define sub-domains of P granules while others (MUT-16) reside outside or at the edge. These observations are intriguing but not conclusive, especially since the terms used (P granule, compartment, nuage) are poorly defined. Also it is not clear what these observations add to the literature given that it was already known that these markers do not co-localize in nuage, and that nuage is a multi-condensate structure. We recommend the following:

- Define terms. It is not clear that the proteome stain corresponds to "P granules", which traditionally have been defined by PGL-1 staining. The EM studies in the past did not distinguish between P/Z granules and mutator foci and it is likely that the pan-proteome stain does not either. It may be more prudent to use the more generic term nuage to refer to the proteome staining, and use the protein-specific reagents to define condensates within the nuage, as is now the norm (e.g. Ouyang and Seydoux, RNA 2022). Different granules/condensates can have distinct compositions (PGL-1 vs ZNFX-1) yet likely have the same protein density.

We have taken Reviewer #1 and #2's very helpful suggestions to clarify our language. We have replaced 'P granule' and 'perinuclear protein density' with 'germ granule'. We kept the term P granule in certain instances only to help general readers i.e. those not from the *C. elegans* community, when specific references are made for past studies on P granules. We have also kept the term 'perinuclear protein density' in a few places in the first section of the Results. This is because we were determining whether those perinuclear density were in fact germ granules in that section. Furthermore, we have cited Ouyang & Seydoux 2022, a highly relevant review, throughout our manuscript.

- Quantify results to justify conclusions. How often are MUT-16 inside vs at the edge of the pan-proteome stain?

As suggested also by Reviewer #1, we have now included analysis for the localisation of PRG-1, ZNFX-1 and MUT-16 (Fig. 3, now Fig. 3 and 4.). This was carried out by using the 'plot profile' function in Image J to measure the intensity along a line that runs across the signals from protein staining. Intensity was then normalised to the signals from the whole granule, i.e. the plots reflect the distribution of the specific proteins (this is to account for the difference in staining from different antibodies/dyes).

We have decided to show examples of MUT-16 being both inside and outside the granules, rather than carrying out a large-scale quantification of this. It would certainly be interesting to carry out such quantification, perhaps also in the mitosis/meiosis stages. However, this is out of scope for our present study.

- Provide control Airy-scan images of non-expanded samples to 1) demonstrate isotropic expansion in all compartments and axes, and 2) illustrate the advantage of the expansion technique. Expansion factors can differ for different organelles within the same cell (Büttner et al., 2020). The authors should show what controls were done to ensure the gel expansion

was isotropic. This is especially important since nucleus diameter was used to normalize the granule size in Fig. 4.

This issue was also brought up by Reviewer #3. We recognise that different cellular components might expand differently, and the best estimation of errors would have been to carry out pre- and post-expansion measurements of germ granules given that this manuscript focuses on the germ granule. However, we were unable to observe germ granules clearly in pre-expansion images. We have measured the aspect ratio of DAPI staining of pre- and post- expansion images (Fig. S1) to show that in X- and Y- directions, the expansion is isotropic in the nucleus, which is close to where the germ granules are located. Furthermore, in the pachytene region, the chromatin are organised in a way that can be threshold easily on Image J for automatic measurement to avoid manual measurements which can lead to errors.

We have added a section in the Discussion to highlight the need to assess isotropy.

- The authors find that there is an average of 2.5 granules per nucleus in a Z-slice, however this data is not well represented by their images. Most nuclei shown appear to contain more than 2 granules. Additionally, previously published reports of non-expanded germline nuclei also appear to contain more than 2 granules per nucleus in a Z-slice. The authors should comment on why expanded nuclei might appear to have fewer granules than non-expanded.

The purpose of this figure is not to show how many granules there are per nuclei but to show that perinuclear protein densities are germ granules, by showing that they contain DEPS-1. In the expanded sample, signals/materials are more 'spaced out' and therefore in certain Z-slice, one would observe less granules than in an unexpanded sample. This is reflected in the variations of the data.

We have realised our plot is misleading. We now show how the data points in the pre- and post- expansion sample are paired to each other to reflect the co-staining of pan-protein stains and GFP-DEP-1 signals.

- Fig. 3B: NHS ester staining does not appear to clearly define the boundary of the P granule. As seen in several examples throughout the paper (especially granule 3 and 4 in Fig. 3B), the edges of the perinuclear densities are blurred together with the nuclei. Before coming to conclusions that mutator foci are within P granules, the boundary of the P granule should be better defined.

While the Reviewer is correct that we are unable to distinguish between nuclei and the germ granule at 3x expansion, we are able to distinguish between germ granule from the cytoplasm which is what is important in this figure.

To better illustrate that mutator foci are found both outside and inside the germ granule, we have measured distribution of proteins along a line in all three fluorescent channels (Fig. 3). Mut-16 signals can be seen in some instances where the proteome signal decreases at the cytoplasmic edge.

- Fig. 3C, 3D: Toraason et al., 2021 found that the frequency of ZNFX-1 colocalization with P granules changes depending on relative gonad position. The authors should comment on

what region of the gonad they imaged these granules and how often they observed colocalization vs. non-colocalization.

Our data throughout the manuscript was taken in the pachytene region. To better illustrate the colocalization vs non-colocalization of PRG-1 and ZNFX-1, we have measured the intensity along several regions of the granules (Fig. 4).

- Fig. 4: The authors should include micrographs of wild-type granules as a comparison for these mutants. Based on the presented images, the *mut-16* mutant granules do not appear smaller than wild-type or *deps-1* mutants, even though that's what is shown in the quantification.

We have included wild-type animals in Fig 5. The measurements can also be found in the Supplemental information. Videos 1 and 2 also show the differences in granules between wild type and *deps-1* mutant well. Please also see comments below.

- Fig. 4D: The units for measuring P granule size is not given. If in microns, can such small differences (less than 40 nm) in granule size be accurately measured given the authors stated they have 40 nm resolution in the xy plane? Also, the authors should show the same experiment in non-expanded germlines to illustrate the necessity of using expansion microscopy to measure these differences.

The units are not in microns. It is expressed as a fraction to the diameter of the nucleus (stated in the legend). This is to account for potential small variations between expansion factor in each germline.

- In the text (last paragraph, pg. 7) the authors state that mutations in mutator foci affect DEPS-1 localization. Later in the paragraph they say that *mut-16* mutation causes loss of 22Gs but not P granule protein localization. These statements seem contradictory. Additionally, there is not enough evidence given to show that P granule morphology is dependent on small RNA levels, since loss of *mut-16* also affects DEPS-1 localization (Suen et al., 2020).

We have clarified in the text that the P granule proteins that are not affected by *mut-16* are PGL-1 (Phillips et al) and PRG-1 (Suen et al). It is indeed an intriguing observation that *mut-16* affects DEPS-1 localisation dramatically and PRG-1 depends on intact *deps-1*. Our hypothesis is that RNA plays a role. For example, Barucci et al 2020 shows that removing 22Gs by knocking down *mut-16* can rescue the *prg-1* mutant mortal germline phenotype.

- Fig. 4B: Based on the NHS ester staining, this P granule does not appear disconnected from the nucleus.

We described it as 'infrequently seen as granules almost dissociated from the nuclear membrane'. This was also shown in Fig S1 (which is now Fig S2). We are trying to describe the granules having a very narrow base as opposed to the wide base in wild type animals. Furthermore, an image J automatic profiling is included as suggested by Reviewer #1 which shows clearly that in one of the instances you can see a dip in pan-protein stain channel (NHS ester stain) at the point where the granule is disconnected from the nuclear membrane.

Reviewer #3 (Comments to the Authors (Required)):

Summary:

Suen et al. present the identification of subdomains in adult *C. elegans* germ granules using an advanced expansion microscopy coupled with bulk labeling of the proteome. Based on colocalization study of the Z granule protein, P granule and Mutator foci protein markers, these subdomains are localized within the same perinuclear protein-dense structures. The size and morphology of these perinuclear protein-dense structures are dependent on both small RNA and protein levels. The enhanced expansion microscopy pipeline developed in this study provides an important tool and a significant advance in further understanding of biogenesis, spatiotemporal regulation and function of the membraneless organelles and liquid-like biomolecular condensates.

There are some substantial concerns and minor comments about figures, interpretation and clarity of writing that would need to be addressed.

Essential revisions:

To the best of my knowledge, this is the first reported implementation of the expansion microscopy workflow to visualize germ granules; therefore validation of isotropy and benchmarking of the overall distortion associated with the entire protocol is required. The P granule diameter distribution after expansion is not linearly preserved (Fig. 1c), this could potentially distort the relative organization of proteins and relative length measurements. Would it be possible to normalize after-expansion feature measurements and to estimate the error of measurements?

This issue was also brought up by Reviewer #2. We recognise that different cellular components might expand differently, and the best estimation of errors would have been to carry out pre- and post-expansion measurements of germ granules given that this manuscript focuses on the germ granule. However, we were unable to observe germ granules clearly in pre-expansion images. We have therefore measured the aspect ratio of DAPI staining of pre- and post- expansion images (Fig. X) to show that in X- and Y- directions, the expansion is isotropic in the nucleus, which is close to where the germ granules are located. Furthermore, in the pachytene region, the chromatin are organised in a way that can be threshold easily on Image J for automatic measurement.

We have added a section in the Discussion to highlight the need to assess isotropy.

Minor comments:

1. Fig. 1b and Fig.1c need to be referred in the Results section.

We thank the Reviewer for pointing this out. We have corrected this.

2. Would it be possible to add % of the surface overlap between NHS ester staining and GFP-DEPS-1 in P granules (Fig. 2)? Colocalization of GFP-DEPS-1 and PRG-1 in the wild-type animals needs to be shown (Fig. 2).

We were unable to find an image J plugin that calculates the overlap between NHS ester and DEPS-1 staining, although it would be a very good addition. However, we have included profile plots to complement PRG-1, ZNFX-1/MUT-16 colocalisation in Fig 3 and 4 (originally Fig 3). DEPS-1 and PRG-1 colocalization images in wild-type animals can be found in Fig 5.

3. The P granule morphology appears variable in the GFP-reporter strains (GFP-DEPS-1 Fig. 2a vs. GFP-MUT-16 and GFP-ZNZF-1 in Fig. 3a-b). Does this reflect expected background level of the P granule heterogeneity?

Yes, it does but it also depends on the cross section that we imaged. It would have been good to use volume data but this is technically challenging.

4. The P granule boundaries are defined by the local protein density (Fig 3b). Is the colocalization pattern of PRG-1 with GFP-MUT-16 and GFP-ZNZF-1 consistent with DEPS-1 staining?

We have not carried out these experiments due to limitation of the scope of this study, although it would have been a good addition. However, we have included PRG-1 and DEPS-1 co-staining results (Fig. 5).

5. The solid lines outlining P granules (Fig. 3b and 3d) need to be replaced by thin dashed-lines that offer better visibility and ease evaluation of the partially colocalized features.

We have removed the boundary in the granule insert of the profile plots to aid visibility.

6. The authors describe the phenotypic consequence of loss of DEPS-1 and MUT-16 (Fig. 4). It would be informative to have quantification of the average granule number/retention per nucleus in wild-type animals, *deps-1* and *mut-16* mutants included in Fig 4.

We apologise for a typo in the manuscript, originally stating that we were measuring average granule number in these mutants. This has now been corrected to state that we are only measuring germ granule size. The reason for not showing the average granule number is that some of the granules were very small and we were uncertain whether it was a granule or not, especially when PRG-1 staining is lacking in some of the granules/PRG-1 localisation is no longer perinuclear. We decided to show the morphological changes in the granules that we were certain as granules by measuring the size.

Referee Cross-Comments:

Overall, there is a substantial overlap and a very balanced complementarity between reviewers' reports. Both, reviewer #1 and reviewer #2 raised important concerns and questions regarding validation of the new technique used in this study, adult *C. elegans* germ granule biology, accuracy and precision in the terminology, the lack of robust quantitative image analysis and data interpretations.

I strongly agree with their comments, that need to be addressed by the authors; nothing to flag specifically.

January 19, 2023

RE: Life Science Alliance Manuscript #LSA-2022-01650-TR

Prof. John E Ladbury
University of Leeds
Cellular & Molecular Biology
Room 7.08, Miall Building
Leeds, North Yorkshire LS2 9JT
United Kingdom

Dear Dr. Ladbury,

Thank you for submitting your revised manuscript entitled "Expansion microscopy reveals subdomains in *C. elegans* germ granules". We would be happy to publish your paper in Life Science Alliance pending final revisions necessary to meet our formatting guidelines.

- please address the remaining Reviewers' comments
- please add ORCID ID for secondary corresponding author-they should have received instructions on how to do so
- please add the Twitter handle of your host institute/organization as well as your own or/and one of the authors in our system
- please add the author contributions and a conflict of interest statement to the main manuscript text
- please provide an ethics statement for your animal study

A. FINAL FILES:

B. MANUSCRIPT ORGANIZATION AND FORMATTING:

Sincerely,

Reviewer #1 (Comments to the Authors (Required)):

The reviewers have addressed my first critique.

One remaining criticism is that the image in new Fig S1ai looks markedly different from the NHS ester images elsewhere, and it isn't clear that germ granules can be deciphered in the expanded image in S1ai or S1aii.

Reviewer #2 (Comments to the Authors (Required)):

The authors have addressed the reviewers comments. I suggest the authors define their terms "germ granules, P granules etc..." upfront to avoid confusion. Right now these terms seems to be used interchangeably in some places and distinctly in others, which is very confusing. It is not necessary for the authors to use the prior used definitions, even when referring to earlier work. Rather they should define their terms and criteria up front and stick to their conventions throughout the paper. Please also consider a table that would summarize their findings and provide an handy reference for what marker they used to identify each condensate type.

Reviewer #3 (Comments to the Authors (Required)):

The revised manuscript is improved in two key ways.

First, the authors clarified their language and P granule-specific terms in the revised version. Second, they have provided overall satisfying answers to initial review and included assessment of isotropy, improved image analysis and quantification, and added a section in the Discussion to highlight limitations of the study.

1) Figure 5 (b, c): the legend text referring to deps-1 and mut-16 (pk710) is inconsistent with the main text (page 7). It might be a swap in the main text.

3) Figure 5(b) and Figure S2(a) show the same staining. Is this image duplication necessary?

Reviewer #1 (Comments to the Authors (Required)):

The reviewers have addressed my first critique.

One remaining criticism is that the image in new Fig S1ai looks markedly different from the NHS ester images elsewhere, and it isn't clear that germ granules can be deciphered in the expanded image in S1ai or S1aii.

We thank the Reviewer for highlighting this point. It should be pointed out that upon expansion smaller structures become visible and there is a marked improvement in the contrast on some images. This would explain the differences seen by the Reviewer.

Reviewer #2 (Comments to the Authors (Required)):

The authors have addressed the reviewers comments. I suggest the authors define their terms "germ granules, P granules etc..." upfront to avoid confusion. Right now these terms seems to be used interchangeably in some places and distinctly in others, which is very confusing. It is not necessary for the authors to use the prior used definitions, even when referring to earlier work. Rather they should define their terms and criteria up front and stick to their conventions throughout the paper. Please also consider a table that would summarize their findings and provide an handy reference for what marker they used to identify each condensate type.

We are grateful for this Reviewers comments regarding the definitions of the important terms in this work. However, we feel that it should be pointed out (and as noted by Reviewer #3) that the definitions in the *C. elegans* field are very specific and we have followed this field for standard terminology and language. We decided not to extend the manuscript by including a Table as noted by the Reviewer. We felt that this was superfluous to needs since there are only three items that would be included.

Reviewer #3 (Comments to the Authors (Required)):

The revised manuscript is improved in two key ways.

First, the authors clarified their language and P granule-specific terms in the revised version. Second, they have provided overall satisfying answers to initial review and included assessment of isotropy, improved image analysis and quantification, and added a section in the Discussion to highlight limitations of the study.

1) Figure 5 (b, c): the legend text referring to *deps-1* and *mut-16* (pk710) is inconsistent with the main text (page 7). It might be a swap in the main text.

We thank the Reviewer for pointing out this error. The text in the manuscript has been corrected.

3) Figure 5(b) and Figure S2(a) show the same staining. Is this image duplication necessary?

Although we appreciate the point raised by this Reviewer, we decided not to include the zoomed Figure in the main manuscript but include this in Supplementary Material. However to do this without confusing the reader we also included the non-zoomed and zoomed images together. This can be altered at the Editors discretion.

January 26, 2023

RE: Life Science Alliance Manuscript #LSA-2022-01650-TRR

Prof. John E Ladbury
University of Leeds
Cellular & Molecular Biology
Room 7.08, Miall Building
Leeds, North Yorkshire LS2 9JT
United Kingdom

Dear Dr. Ladbury,

Thank you for submitting your Research Article entitled "Expansion microscopy reveals subdomains in *C. elegans* germ granules". It is a pleasure to let you know that your manuscript is now accepted for publication in Life Science Alliance. Congratulations on this interesting work.

DISTRIBUTION OF MATERIALS:

Again, congratulations on a very nice paper. I hope you found the review process to be constructive and are pleased with how the manuscript was handled editorially. We look forward to future exciting submissions from your lab.

Sincerely,
